



# Evaporating brine from frost flowers with electron microscopy, and implications for atmospheric chemistry and sea-salt aerosol formation

Xin Yang[1], Vilém Neděla[2], Jiří Runštuk[2], Gabriela Ondrušková[3], Ján Krausko[3], Ľubica Vetráková[3], Dominik Heger[3]

[1] British Antarctic Survey, Natural Environment Research Council, Cambridge, UK
[2] Environmental Electron Microscopy Group, Institute of Scientific Instruments of the CAS, Brno, Czech Republic
[3] Department of Chemistry, Faculty of Science, Masaryk University, Kamenice 5/A8, 625 00 Brno Research, and Centre for Toxic Compounds in the Environment (RECETOX), Masaryk University, Kamenice 5/A29, 625 00 Brno

*Correspondence to*: Xin Yang (xinyang55@bas.ac.uk), Dominik Heger (hegerd@chemi.muni.cz)

**Abstract.** An environmental scanning electron microscope was used for the first time to obtain well-resolved images, in both temporal and spatial dimensions, of lab-prepared frost flowers (FFs) under evaporation within the chamber temperature range from -5°C to -18°C and pressures above 500 Pa. Our scanning shows temperature-dependent NaCl speciation: the brine covering the ice was observed at all conditions, whereas the NaCl crystals were formed at temperatures below -10 °C as the brine oversaturation was achieved. Finger-like ice structures covered by the brine, with a diameter of several micrometres and length of tens to one hundred micrometres, are exposed to the ambient air. The brine-covered fingers are highly flexible and cohesive. The exposure of the liquid brine on the micrometric fingers indicates a significant increase in the brine surface area compared to that of the flat ice surface at high temperatures, whereas the NaCl crystals can become sites of heterogeneous reactivity at lower temperatures. There is no evidence that, without external forces, salty FFs could automatically fall apart to create a number of sub-particles at the scale of micrometres as the exposed brine fingers seem cohesive and hard to break in the middle. The fingers tend to combine together to form large spheres and then join back to the mother body, eventually forming a large chunk of salt after complete dehydration. A present microscopic observation rationalizes several previously unexplained observations, namely, that FFs are not a direct source of sea salt aerosols and that saline ice crystals under evaporation could accelerate the heterogeneous reactions of bromine liberation.

## 1 Introduction

Ice and snow constitute an important reaction medium not only on the Earth but also in the outer space, and both are known to accumulate and concentrate significant amounts of impurities that are stored, transformed, and eventually released. The knowledge of the exact location and speciation of these chemical impurities in ice and snow under various environmental conditions is crucial for assessing the reactivity (McNeill et al., 2012;Bartels-Rausch et al., 2014;Gudipati et al., 2015) and further fate.

The ions originating from sea salt (including, for example, Na$^+$, Cl$^-$, and Br$^-$) have been widely observed in polar regions in media such as aerosols, snow packs, and ice cores (DeAngelis et al., 1997;Rankin and Wolff, 2003;Fischer et al., 2007;Legrand et al., 2016 ). The sea salts trapped in snow packs form a large chemical reservoir and therefore embody a significant part of chemical reactions in the polar boundary layer (Abbatt et al., 2012). Conversely, inactive ions such as Na$^+$ recorded in ice cores could serve as a climate index for the past climate (Rankin and Wolff, 2003;Abram et al., 2013). Although the sea spray and bubble bursting in the open ocean surface dominate sea salt aerosol (SSA) production in most of the Earth, the winter SSA peaks observed at most near-coastal sites (Wagenbach et al., 1998;Rankin et al., 2004) are clearly out of phase with the distance to the open water. Several lines of evidence suggest that winter sea salt




cannot derive only from the long-range transport of the aerosol produced over the open ocean. The winter maximum observed seems inconsistent with the fact that the nearest open water is hundreds of kilometres further away in the given season because of extended sea ice. In ice cores, significantly higher concentrations of salts are found in glacial periods, when sea ice was even more widespread, and furthermore when relevant models do not suggest any greater transport

(Mahowald et al., 2006). The most direct evidence of the salt that should originate from zones covered with sea ice arises from the composition of sea salt aerosol and ice cores. Frequent episodes when the sulfate/sodium $[SO_4^{2-}/Na^+]$ ratio is below that of seawater, despite the addition of the non-sea-salt sulfate resulting from the oxidation of dimethlysulfide, are observed (Wagenbach et al., 1998). This is believed to occur due to the effect of mirabilite ($Na_2SO_4.10H_2O$) precipitating from the brine when the temperature drops below -8°C (Wagenbach et al., 1998;Jourdain et al., 2008 ), a segregation

inapplicable to sea spray particles.

Frost flowers (FF) are commonly observed on fresh sea ice and preferentially grow on small-scale roughness nodules sticking above the surface or out of the brine, which is typically colder by 5 °C compared to bulk ice (Domine, 2005;Galley et al., 2015); at these conditions, the supersaturation of water vapour is frequently achieved (Style and Worster, 2009). Frost flowers often consist of featherlike dendritic ice crystal structures, and their surface can be covered

by concentrated brine (Perovich and Richter-Menge, 1994;Barber et al., 2014;Galley et al., 2015). A detailed chemical composition analysis was performed, finding, inter alia, that FFs can reach the salinity of the concentrated brine of 120 practical salinity units (Douglas et al., 2012). FFs have the specific surface area of 185 (+80-50) cm²/g, measured by methane adsorption; such a specific surface area is about five times lower than that of freshly fallen snow. The surface area of frost flowers is estimated to be 1.4 m² per m² of ice surface (Domine, 2005). The fragile structure plus extremely

high salinity (Rankin, 2002) make FFs the medium probably causing the chemical reaction (Perovich and Richter-Menge, 1994;Kaleschke et al., 2004;Simpson et al., 2007) and embodying the SSA source (Wagenbach et al., 1998;Wolff et al., 2003). However, recent studies propose that FFs are not as important as assumed previously (Obbard et al., 2009;Roscoe et al., 2011;Abbatt et al., 2012). In particular, a recent wind tunnel experiment indicated that FFs are not a direct source of SSA (Roscoe et al., 2011). Apart from saline FFs, the snow lying on sea ice was also hypothesized to be an efficient

source of SSA and bromine via blowing snow events (Yang et al., 2008;Legrand et al., 2016;Zhao et al., 2016).

In any case, the formation of SSA from salty ice particles requires its size to be reduced via the loss of water through either evaporation or the sublimation process, depending on the temperature. Until now, there has been no detailed image at the micro-physical scale to indicate what happens to saline ice under the evaporation or sublimation processes.

Moreover, current atmospheric chemical models consider the solutes' impurities on ice to be present in a diluted

liquid solution on the ice surface (Domine et al., 2013). Such a model is generally unsatisfactory in describing the real situation, and thus more realistic parameters for modelling are needed. Some of us previously showed that the concentration increase of nonpolar (Heger et al., 2011;Kania et al., 2014;Krausko et al., 2015a;Krausko et al., 2015b ) and polar compounds (Heger et al., 2005;Heger et al., 2006;Heger and Klan, 2007;Krausková et al., 2016) can even lead to their crystallization under certain conditions.

In this study, we grew FFs in a laboratory and inspected them using an environmental scanning electron microscope (ESEM) to obtain some information about the state of impurities in/on the ice. The preparation of the FF samples to mimic the FFs naturally produced on sea ice is detailed in section 2. The information related to the ESEM is in section 3. The scanning results are presented in section 4; the atmospheric impact is discussed in section 5; and the conclusions are available in section 6.






## 2 Growth of the frost flowers and preparation of the samples

The FFs were prepared in a custom-built 2m × 2m walk-in cold chamber. Inspired by the natural condition at which FFs grow (Style and Worster, 2009) and exploiting previous methods of preparation (Roscoe et al., 2011), we cooled the walk-in cold chamber down to -30°C and inserted vessels containing pure water or an aqueous solution of NaCl (3.5% w/w, similar to that of sea water) at 20°C. The vessels were isolated with Styrofoam to minimize the contact cooling of the solution by the floor of the walk-in chamber and to promote cooling by the air. We typically observed the following course of events: First, hoarfrost appeared on the sides of the beaker; then, an ice crust formed on the water level; and, subsequently, dendrite-shape icy features (considered to be FFs) grew gradually, as shown in Figure 1. After the ice reached a certain thickness, the FFs stopped growing and were collected into a pre-cooled vial to be stored at the temperature of liquid nitrogen. Care was taken to collect only the FFs from the ice surface, avoiding the hoarfrost condensed on the walls of the beaker. The FFs were fragile and fragmented during the manipulation. The FFs grown on the surface of pure water were powdery; however, those grown from the brine were extremely sticky, and therefore two spatulas were needed to place them into the vials. We attempted to follow growth conditions similar to the natural ones; our sampling method guarantees that the features were grown on the ice surface, and thus the examined samples are believed to be very similar to natural FFs.

## 3 Environmental scanning electron microscope (ESEM)

The ESEM (AQUASEM II) is unique in the observation of nonconductive, wet, or liquid samples, with the specimen chamber pressure as high as 2000 Pa and temperatures ranging from 0°C to –30 °C (Tihlarikova et al., 2013). The indicated temperature is measured on the sample holder. The temperature of the ice surface is estimated not to differ by more than 2°C from that of the holder on which the temperature is measured. This estimate is based on the observation of the ice surface melting. The major source of the heat is the energy from the electrons used for scanning.

The conditions inside the chamber allow for the observation of ice samples in conditions similar to those under which ice and snow occur naturally. No conductive coating of the sample is needed, because the positive ions resulting from the electron-gas ionisation in high gas pressure conditions of the ESEM discharge the accumulated charge. The strength of this apparatus lies in the delicate control of the dynamic conditions in the specimen chamber via an originally designed hydration system enhanced with temperature and vapour flow control and an advanced cooling system integrated in the sample holder. The specimen chamber can be evacuated very slowly, with the possibility of reaching high humidity conditions in the sample vicinity without purge-flood cycles (Neděla et al., 2015). The water vapour temperature is estimated to be around 10°C. Care was taken to direct the steam away from the sample to prevent any heat-up. The regulation of the temperature in the vicinity of the sample allows us to study ice in precisely controlled conditions (Krausko et al., 2014). The temperature, pressure, and relative humidity in the chamber of the ESEM can be set close to the frost point to cause ice sublimation or gradual growth. The ESEM is equipped with a tungsten hairpin cathode as a source of electrons and also with two custom built detectors (Neděla et al., 2011): an ionization detector for secondary electrons (surface sensitive to provide information about the morphology of the ice surface), and a highly material sensitive detector of back-scattered electrons. A comparison of these two modes on identical samples yields complementary information on the morphology of the ice surface and ice grain boundaries contaminated by impurities.

As shown in Figure 2, water vapour flows around the sample and through the detector's aperture during the scanning of the sample. The flow speed varies from 2 m/s on the sample surface to 16 m/s at the distance of 0.7 mm above the sample surface (simulated for the experimental pressure of 300 Pa in the specimen chamber of the ESEM AQUASEM




II and for the spherical shape of the sample). The flow is influenced by the shape of the sample, pumping speed, and ESEM aperture diameter. The flow speed was simulated as described previously (Maxa, 2011;Maxa, 2016).

**4 Results**

**4.1 FFs at a high temperature: brine fingers formation**

The FFs were scanned at the chamber temperature of -5.2°C. Figure 3 shows many spikes sticking out from the main ice body; here, these will be referred to as *fingers*. The smooth texture is indicative of surfaces covered with a layer of a solution in contrast to the dry ice crystal surface observed at temperatures below -30°C and pressures below 50 Pa (McCarthy et al., 2007;Blackford, 2007;Pfalzgraff et al., 2010;Bartels-Rausch et al., 2014). The image differs from that of a water drop also in the irregular and non-spherical features. Thus, we are of the opinion that the exposed finger-like spikes consist of ice covered with a brine. More arguments towards this interpretation will be proposed in the following parts of the text. The brine is expected to become more concentrated as a result of the loss of water during progressive evaporation. The exposed thin fingers can be as much as one hundred micrometres long, still remaining quite cohesive and hard to break. In Figure 3f, we estimate the thickness of the fingers' necks at their most narrow points to be $d$ = (2.23 ± 0.43) µm; the given values are (mean ± standard error of the mean). In some cases, a rounded sphere appeared on the top of a finger during evaporation, as encircled in Figure 3. At temperatures exceeding ~ -10°C, which is well above the eutectic point temperature ($T_{Eutectic}$= -21.21 °C) (Brady, 2009), the concentrated brine was always observed as liquid, and no NaCl crystals were perceived. The viscosity of the concentrated brine at the ambient temperature is less than two times that of pure water. (Weast et al., 1987) Although we did not find any reference to the values of the brine viscosity at sub-zero temperatures, the viscosity of sea water at zero temperature is only slightly higher than that of pure water (1.3 times) (Sharqawy et al., 2010), and the viscosity of supercooled water at –17 °C is only 3.8 times larger compared to that at 20°C (Dehaoui et al., 2015). Therefore, we do not assume that the viscosity of the brine will increase significantly enough to be the only explanation for the formation of the fingers. The fingers were observed to easily bend and flap following the airflow in the chamber (Figure 3, oval and S1). When these fingers are close enough to one another, they may tangle together to join into a larger one.

The relative humidity in our experiments was set to be slightly below the frost point, and therefore slow loss of the water from the sample could be observed. Thus, the micrographs obtained already at the beginning of the observations are not fully undisturbed; we assume that the water evaporates faster from the brine of a lower concentration compared to the more concentrated one (in accordance with Raoult's law). The vapour pressures above the water, ice, and saturated brine (8.3 % w/w) at -5°C are 422, 402, and 403 Pa, respectively. The applied equations for the vapour pressure above the water and ice are adopted from Buck (Buck, 1981); for the brine, the relevant formulae are proposed within the article by Perovich and Richter-Menge (Perovich and Richter-Menge, 1994):

$$e_w = [1.0007 + 3.46 * 10^{-6}p] * \left[6.1121 * e^{\left(\frac{17.966*t}{247.15+t}\right)}\right]$$

$$e_i = [1.0003 + 4.18 * 10^{-6}p] * \left[6.1115 * e^{\left(\frac{22.452*t}{272.55+t}\right)}\right]$$

$$e_b = e_w(1 - 0.000537 \times S_b)$$





where $e_w$ is the saturation vapour pressure above the water, $e_i$ is the saturation vapour pressure above the ice, $e_b$ is the saturation vapour pressure above the brine, $p$ is the atmospheric pressure in millibars, $S_b$ is the brine salinity in parts of mass per thousand, and $t$ is the temperature in °C.

In an additional experiment with pure water FFs (not shown here), we found out that these sublimate markedly faster than brine-covered FFs.

At the temperature of -5.2 °C and concentration of NaCl lower than 23.3 % (w/w), the phase diagram (Figure 4) indicates the presence of a liquid solution of NaCl and ice. Therefore, if the equilibrium conditions are established, there will be ice and ca. 8.3 % NaCl solution covering its surface. As the water is gradually evaporated from the brine, the ice must melt to maintain the equilibrium concentration. This process is represented with the red arrow in the phase diagram of Figure 4. This rationalizes well our observations: the evaporation of the water from the brine on the fingers causes its concentration to increase above the equilibrium concentration; therefore, the water must be supplied from the ice body towards the brine fingers to dilute the brine. This process results in gradual melting of the ice body till all the ice is melted.

An examination of the sequences of the micrographs suggests that the evaporation proceeds faster from the main ice body than from the fingers. This can be seen in the video of S1 as the fingers exhibit a relatively stable shape even if the main ice body gradually abates. Thus, the concentration of the brine in the surface layer of the fingers is deemed to be higher than that on the main body. We can speculate that the higher concentration of NaCl on the fingers is a result of previous water vapour evaporation from the brine on the fingers. Possibly, the most concentrated solution is found on the tips of the fingers, where small spheres are sometimes formed. The increased local concentration of salt would effectively lower the water vapour evaporation and hence reduce further melting of the ice forming the fingers' interior, thus not allowing its breakaway from the main body. For example, if the NaCl saturation concentration of 25 % (w/w) is reached at -5°C, the water partial pressure drops to 365 Pa from the 403 Pa at the brine equilibrium concentration (8.3 %).

A particular consequence of a higher rate of water evaporation from the side-wall of a finger and the main ice body compared to the fingertip is the formation and propagation of gulfs. This is well exemplified in Figure 5, where the process resulted in the breakaway of large pieces of ice (>100 micrometers) from the mother body. First, a very deep gulf was formed which later separated the two pieces by a very thin neck, eventually leading to the breaking off of the two parts (S2). This case indicates that the evaporation or sublimation process indeed could cause a large ice particle to fall aside, but this phenomenon is not common, as we noticed it only once in all our observations (twenty experiments). Moreover, there is no evidence that the brine fingers can fall apart to form a number of micrometer-sized particles.

The understanding of the structure of FFs is still far from complete. The 3D X-ray micro computer tomography experiments suggest that salt impurities are present mostly on the ice surface (Hutterli et al., 2008). Such a finding is consistent with our observation and can be well understood, taking into account the genesis of FFs, where the brine wicks on the already formed ice to develop a highly saline surface skim (Domine, 2005). By contrast, the dynamics of freezing forces the solutes to segregate and form the veins of freeze concentrated solutions engulfed by the ice (Blackford, 2007;Cheng et al., 2010;McCarthy et al., 2013;Bogdan et al., 2014;Krausko et al., 2014 ). The solutes in the freeze-concentrated solution, here probably in the surface layer only, experience, besides an increased concentration (Heger et al., 2005;Kania et al., 2014;Krausko et al., 2015a), also a changed pH (Heger et al., 2006;Krausková et al., 2016) and polarity (Heger and Klan, 2007). Recently, it was noticed that, for a frozen solution, the surface brine layer is interconnected with the interior veins system (Walker et al., 2013).

It is interesting to note that an anomalous increase of the water heat capacity with a decreasing temperature is reduced and even eliminated with the increasing salt concentration of the solution. The isobaric heat capacity $c_p$ of 23 %

brine at -17.2°C equals ca. 3.3 kJ°C⁻¹kg⁻¹ (Archer and Carter, 2000), which is substantially less compared to water at same temperature (4.3 kJ°C⁻¹kg⁻¹). Conversely, the $c_p$ of ice is still much lower than that of water or brine (1.98 kJ°C⁻¹kg⁻¹ at -17.2°C) (Haida et al., 1974). Therefore, at thermal gradients, ice will change its temperature faster than the liquid parts of the system.

Overall, ice covered with brine seems to offer the most reasonable explanation for the objects denoted as fingers, aptly characterizing all our observations and also corresponding to the previous studies (Domine, 2005;Cheng et al., 2010). The observed generation of fingers can be the energetically most feasible path to deal with a large amount of concentrated brine being relatively quickly formed on an ice body, whose volume gradually decreases. Further water
evaporation would concentrate the brine, eventually forming NaCl crystals. The final product of the evaporation is shown in Figure S3. Typically, a sample was dried within 30 minutes in the microscopic chamber (depending on the exact experimental condition), the relative humidity embodying the most important factor (Neděla et al., 2015).

### 4.2  FFs at a low temperature: NaCl crystal formation

The evaporation of FFs at the temperature of -17°C was also scanned, as shown in Figures 6-9. Additionally to the liquid brine observed at -5°C, we saw a large number of salt crystals widely spread on not only the ice surface layer but also the surfaces of the fingers (Figure 6b). Apparently, as we observed the salt crystals, brine, and ice together at these conditions, the sample cannot be in the thermodynamic equilibrium.

The fingers at -17°C were less numerous and more robust compared to those observed at higher temperatures, with their necks typically reaching tenths of micrometres or more (Figure 6). The flexibility of the fingers is demonstrated in Figure 7 by the observation of thin neck tethering and eventual pulling back a large piece of ice to the main ice body. Together with our additional scanning performed at the temperatures of -10°C and -12°C (not shown), we witnessed that chamber temperatures progressively decreasing below -10°C effectively increase the formation of crystals and reduce the
number of fingers, indicating that the brine microphysical feature under sublimation or the evaporation process is temperature sensitive. Figure 8 clearly shows that salt crystals are widely formed on the surface brine layer during water evaporation from the FFs, eventually growing into a large cluster of crystals covering most of the surface. The dynamics of the process is shown by successive images joined into the video in S5. The size of the crystals varies from a few micrometres at an early stage to more than one hundred micrometres at a later stage. It is also possible to discern that salt
crystals freely move on the brine surface, occasionally sinking below the surface. This can provide some indication of the thickness of the brine on the ice surface. It should be noted that, at the temperature of -17°C, air with the relative humidity of ca. 20 % was always used instead of the pure water vapour applied at -5°C, thus setting slightly evaporative conditions in the microscopic chamber.

We suppose that the sample heated up from the liquid nitrogen temperature to -17°C would allow the brine layer
to approach the thermodynamic equilibrium concentration of 20 % (w/w). Referring to the mirabilite, fast dissolution of hydrohalite crystals is expected upon warming (Butler et al., 2016); therefore, the observations are dependent on the temperature and pressure in the microscope's chamber and not on the thermal history of the sample. Further water evaporation can easily cause the oversaturation of the brine solution to a concentration exceeding 24 % (w/w) and thus result in the consequent formation of salt crystals. Presumably, this occurs in our microscopic chamber at the above-
indicated observation temperatures. We can exclude the assumption that the formed crystals are made of water ice as they grow (in size and number) during the evaporation process. The saturation can be reached via increasing the brine concentration by only 3 %, which can easily happen. This process is represented by the green arrow in the phase diagram



of Figure 4. The formation of salt crystals, besides ice melting, is apparently the second mechanism of reducing the brine concentration. Which mechanism prevails then depends on the subtle balance of the vapour pressure and temperature in the microscopic chamber. Under thermodynamic conditions, the crystallization would effectively reduce the salt concentration of the brine to the quasi-equilibrium state of ca. 24%, and a further decrease would occur by the ice melting to 20 %, which still seems to play an important role even at this temperature. For the above considerations, we deliberately separated the three-phase system to two systems in two phases, ice with brine and brine with NaCl crystals, to estimate the equilibrium conditions. The NaCl crystallization heat is slightly exothermic (-3.9 kJ/mol) (Sanahuja and Cesari, 1984), and therefore the crystallization process also supplies some heat for further water evaporation.

We should admit that we do not have any reliable method to decide what kinds of salt crystals are formed in our observations, namely, whether they are anhydrous NaCl (halite) or dihydrate $NaCl.2H_2O$ (hydrohalite). However, the prevailing morphological shapes lead us to prefer the presence of NaCl. The binary phase diagram for water-NaCl (Figure 4) suggests the stability region of NaCl at temperatures higher than 0.11°C; below this temperature, only $NaCl.2H_2O$ is stable. Halite crystallizes in the cubic crystal structure, whereas hydrohalite does so in the monoclinic one. The hydrohalite crystals rapidly recrystallize to anhydrous halite and brine at temperatures of >0.11°C; the reverse recrystallization of halite to hydrohalite is slow even in contact with a saturated sodium chloride solution (Bode et al., 2015).

In aerosol simulating chambers under the conditions of preferential homogenous nucleation, the formed crystal structures do not correspond to those of the phase diagram. Halite crystals were observed at the temperature where the bulk phase diagram predicts the formation of hydrohalite. Only below a certain temperature (varying in two independent experiments: -38.2°C (Wagner et al., 2012) and -21.2°C (Wise et al., 2012)), the efflorescence of hydrohalite crystals resulted from homogeneous crystallization at a specified relative humidity. On the other hand, heterogeneous nucleation on available surfaces, such as ice surface, resulted in the growth of thermodynamically stable hydrohalite. Hydrohalite was found to crystallize from an oversaturated aqueous solution (brine) below the temperature of -0.1°C (Light et al., 2009;Light et al., 2003).

According to the bulk state diagram for the sodium chloride-water system, both the formation of $NaCl.2H_2O$ at sub-zero temperatures and concentration not exceeding 61.9 % should occur. Even though we cannot estimate the oversaturation in the brine surface layer, we do not suppose that the water sublimation from the brine is rapid enough to increase the concentration above 61 %; respecting this argument, $NaCl.2H_2O$ hydrohalite crystals should be formed. Conversely, the shape of the most (but not all) of the salt crystals is close to rectangular, and therefore the cubic structure of halite can be inferred. The variety of NaCl crystal morphologies is presented in Figure 9. It can be argued that, similarly to the non-thermodynamic homogeneous crystallization in aerosol chambers, halite preferentially crystallizes also in the conditions of our observation, for reasons we are currently unable to explain.

## 5 Atmospheric implications

Although our laboratory-prepared FFs can be regarded as one particular example of the natural FF, we cannot determine how representative this concrete example is, especially as the FF's interior structure has not been detailed yet. However, we can surely consider our observations a good model for the general case of the ice-NaCl system. We should stress that our observations were performed at the chamber pressure of $p$ = ~600 Pa, which is substantially lower compared to low atmospheric conditions; therefore, direct implications for the natural FF should be made with care and questioned in further work. Nevertheless, our observations reveal some possibly relevant facts, and these are as follows:



### 5.1 In atmospheric chemistry

Exposing a progressively concentrated brine to the ambient air, following the evaporation of water, may have a significant atmospheric implication, especially in atmospheric chemistry. Depending on the original location of the brine, namely, if it was placed on the ice surface or buried in-between the ice crystals in the vein channels and pockets, the evaporation of the surrounding ice may increase the brine surface area by several times or even more than an order of magnitude. This could potentially accelerate the heterogeneous reactions; one particularly important reaction is bromide liberation, $HOBr(g)+Br^- \rightarrow Br_2(g)$, which is believed to be the direct source of bromine from the saline particles in polar regions (Fan and Jacob, 1992).

As reflected in the images taken, the exposed brine fingers may tangle together and combine with the mother body to form a large chunk of salt in the end (Figure S3). Therefore, the increase in the surface area of the brine due to the exposure of the brine fingers to the ambient air could only be efficient during the evaporating period as aged FFs may exhibit a reduced area due to the formation of a precipitate. Thus, the acceleration of heterogeneous chemistry due to the evaporation process likely applies to fresh FFs and salty snows but not aged ones.

Compared to the crystals lying on the sea ice surface, those aloft snow particles may be more prone to losing their water (Mann et al., 2000). Therefore, salty blowing snow particles lofted from the surface may suffer from rapid loss of water and enhance bromide liberation, as reflected in recent measurements (Jacobi et al., 2012;Lieb-Lappen and Obbard, 2015). Note that the effects of air ventilation in snow packs on snow chemistry, via the above mentioned sublimation process, remain unknown to date and thus deserve further in situ measurement.

Even though atmospheric conditions on the Earth do not often allow for the formation of hydrohalite from bulk brine or in aerosols (Koop et al., 2000;Cziczo and Abbatt, 2000;Wagner and Mohler, 2013), we demonstrate that the local concentration on ice covered with brine exposed to desiccation by wind ventilation can easily meet these conditions. In the real world, an extremely dry condition is not common in most sea ice-covered zones; however, the wind ventilation effect could also cause ice water loss even under a high relative humidity condition (Thorpe and Mason, 1966). The ventilation effect is efficient and could dominate the water loss in the chamber even when the relative humidity is close to 100%. Under the Earth's atmospheric conditions, it could be possible that the ventilation effect is strong enough to trigger NaCl crystal formation. However, anhydrous crystals are not easily prevented from deliquescing; most likely, these crystals will be soon diluted to form brine again. From the general point of view, the impact of the temperature and ventilation effects on saline brine microphysical features, as observed in this study, is interesting and may have significant implications for atmospheric chemistry and climate, for example, with respect to ice nucleation (Wagner and Mohler, 2013).

### 5.2 In sea salt aerosol formation

It seems that, without external forces such as collisions or wind cropping, evaporation itself will not automatically cause particle splitting to form sub-particles. The sticky brine fingers tend to combine back to the mother body (as shown in Figure 7) rather than to fly away to form sub-particles. The leftover of FFs' evaporation is normally a large chunk of salt, as shown in Figure S3. Thus, the present study supplies a clear micro-physical picture in the explanation of why FFs could not be a direct source of SSAs, which is in accordance with the observation by Roscoe et al. (Roscoe et al., 2011) that no SSAs were detected at wind tunnel speeds up to 12 m/s.





Regarding less salty snow particles, it is not clear whether the sublimation process will cause splitting. There is always a potential for large snow particles, e.g., ones with the size of hundreds of micrometres, to split during the evaporation process, especially when the surface brine skim is discontinuous. However, in much smaller particles (tens of micrometres or less), the splitting is less likely compared to the larger ones. Although the results of this study indicate

that the ratio (of the number of SSAs formed from one snow particle) could be close to the order of units, the dependences of the ratio on the particle initial size and salt content are not known. In the original formula for the parameterization of SSA production from blown snow (Yang et al., 2008), the ratio is assumed to be a unit; however, the large number of five was applied in a recent model integration (Huang and Jaeglé, 2016).

The SSA produced from blowing snow forms a reservoir of various chemical compounds; moreover, they become chemically active once they are airborne. For example, the SSA produced can get acidified quickly by absorbing naturally generated or anthropogenic sulphate or nitrate gases, which is a key step for bromide liberation from saline particle (Abbatt et al., 2012). Model integrations with this SSA produced from blowing snow as a source of bromine can aptly capture the observed bromine explosion and ozone depletion events often occurring in polar spring time (Yang et

al., 2010;Legrand et al., 2016;Zhao et al., 2016).

### 6 Conclusions

An ESEM was used, for the first time, to obtain a detailed microphysical picture of evaporating saline frost flowers. The thorough scanning, in both temporal and spatial dimensions, reveals a secret world of FFs in their evaporation period.

The evaporation of water from the brine causes ice melting underneath as it supplies the melting water to dilute

the locally increased salt concentration. This process results in the formation of naked fingers standing out of the main body of the FFs. These fingers covered with the concentrated brine supply an enhanced surface area where (heterogeneous) reactions, exemplified by bromide release, could be boosted. This microphysical picture applies to not only the high saline FFs but also the less saline snows, including blown ones.

The exposed brine fingers are rather sticky and flexible at a higher temperature (e.g., -5°C); they, however,

become stiff with the temperature dropping, due to a lower amount of liquid in the brine. A multitude of micrometric NaCl crystals were observed in the brine layer at temperatures below -10°C, indicating that the brine's microphysical feature is temperature sensitive, thus changing the physical and optical properties of the FFs. As a newly discovered aspect, the presence of NaCl crystals should be considered with respect to possible atmospheric heterogeneous reactivity and the distribution of ions in bulk ice.

It is very likely that, without external forces, the evaporation process itself will not automatically cause a saline crystal to fall apart to produce aerosol size particles. The sticky brine fingers tend to tangle each other and eventually unite with the main body instead of forming multi sub-particles, indicating that FFs are not a direct source of SSA, which is consistent with previous suggestions (Roscoe et al., 2011). This technique allows us to observe liquid NaCl brine on the ice surface and the process of its evaporation.


### Acknowledgements

This work was supported by the BAS Collaboration Fund, UK and Czech Science Foundation (GA14-22777S, GA15-12386S); the affiliated RECETOX research infrastructure is funded by projects of the Czech Ministry of Education (LO1214), (LM2011028).



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

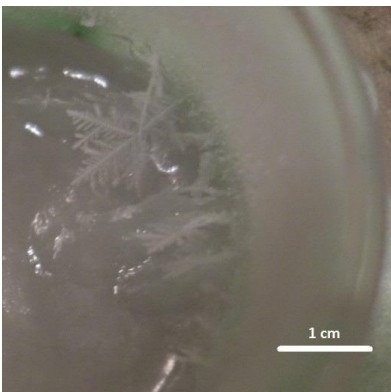

**Figure 1: A frost flower grown in a polystyrene-isolated beaker in a walk-in cold room, at the temperature of -30°C. Both pure**
**water FFs and saline FFs were prepared for further microscopic scanning (see the text for details).**





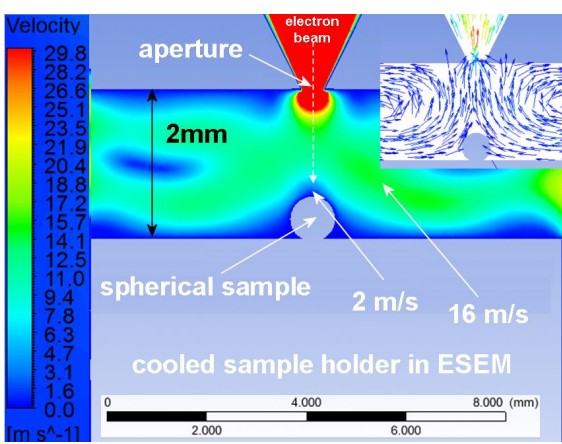

**Figure 2: The ANSYS-based simulation of the water vapour velocity distribution, and the direction of the vapour flow in the vicinity of the sample surface in the specimen chamber of the applied ESEM AQUASEM II.**

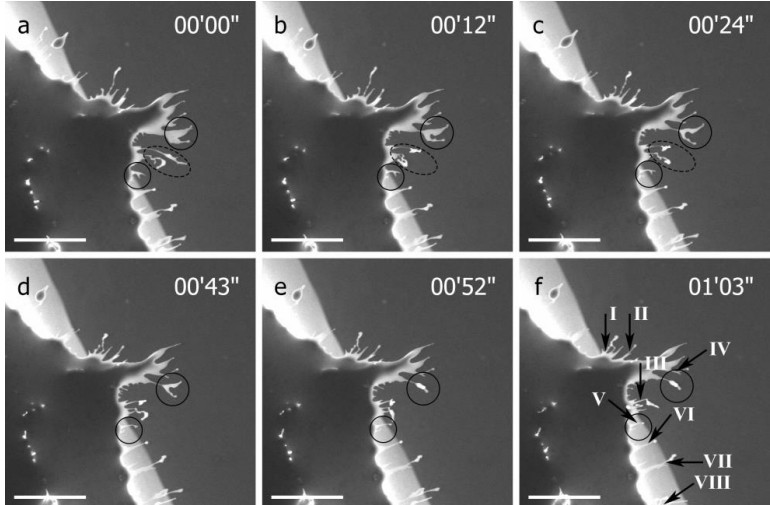

**Figure 3. The dynamical in-situ images of the formation of brine fingers during slow evaporation of water from the FF. The individual fingers bending and flapping around are highlighted in circles. The width of the seven indicated necks in Fig. 3f is measured as $d_1 = (2.23 \pm 0.43)$ µm, mean ± standard error of the mean. Imaged with the applied ESEM AQUASEM II; beam energy 20 keV, ionisation detector, water vapour pressure 348 Pa, sample holder temperature -5.2°C, sample to aperture distance 2 mm. Scale bar: 100 µm. A video of this case is attached (S1).**





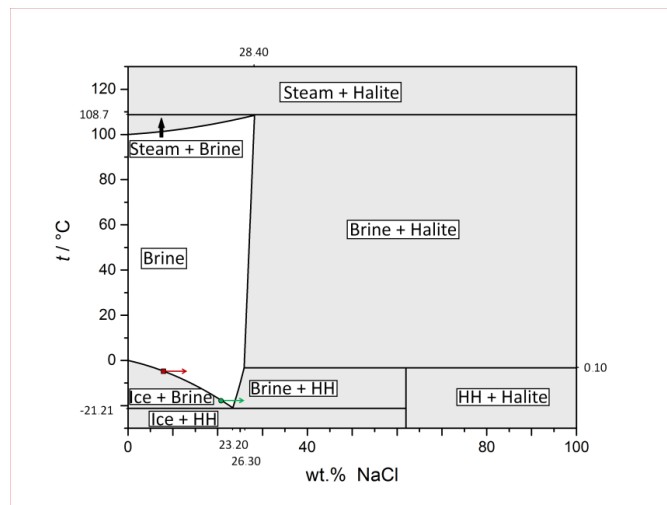

**Figure 4: The phase diagram for a water/NaCl system. Indicated (red and green arrows) are our experimental conditions at about -5°C and -17 °C. HH stands for hydrohalite (NaCl.2H$_2$O). Based on equations from (Brady, 2009).**

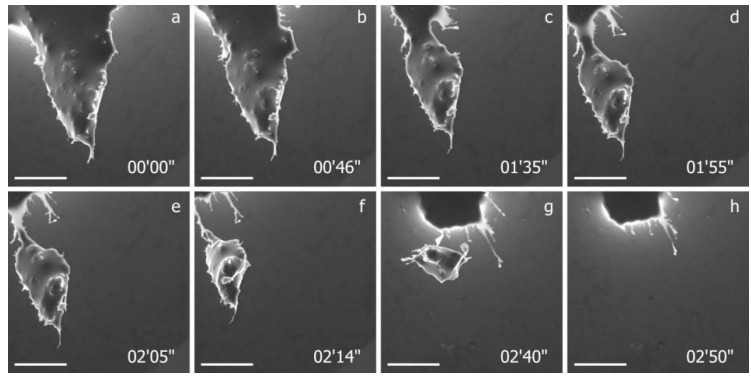

**Figure 5. The dynamical in-situ micrographs of a large size (~100 μm), brine-covered piece of ice formation and breakaway during slow evaporation of water from the FF. Imaged with the ESEM AQUASEM II; beam energy 20 keV, ionisation detector, water vapour pressure 348 Pa, sample holder temperature -5.2°C, sample to aperture distance 2mm. Scale bars: 100μm. A video of this case is attached (S2).**

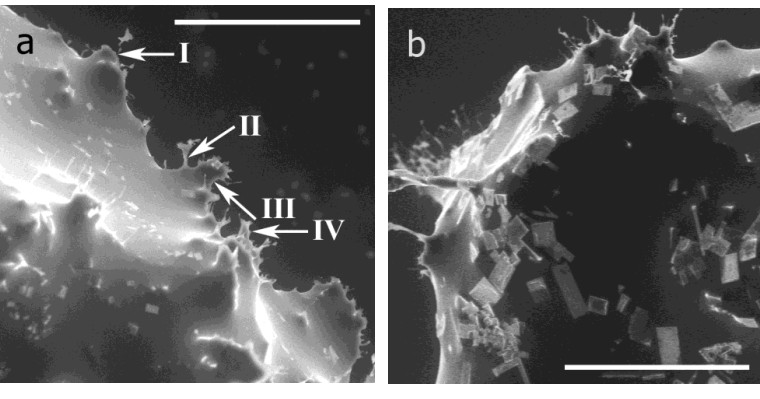





**Figure 6 a, b: The FF micrograph detailing the surface scattered by the NaCl crystals and the finger structures. Compared to the situation at a higher temperature (Fig. 3 and 4), the brine fingers looked stiffer, appeared more rarely, and the angle at the base was larger. The widths of the four indicated necks measured are $d_I$ = 0.87 µm, $d_{II}$ = 3.78 µm, $d_{III}$ = 26.30 µm, and $d_{VI}$ = 13.60 µm. Scale bars: 200µm. Figure 6b shows that the salt crystals can be found also on the protruding fingers. Microscopic conditions: the ESEM AQUASEM II, ionisation detector, air pressure 520 Pa, sample holder temperature -17.0°C, sample to aperture distance 2 mm.**

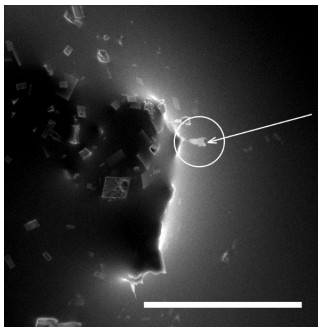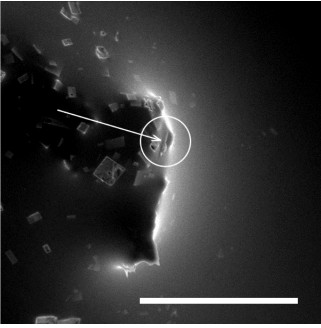

**Figure 7: The FF micrograph showing a finger combining back to its mother body (circled). The ESEM AQUASEM II, ionisation detector, air pressure 520 Pa, sample holder temperature -17.0°C, sample to aperture distance 2 mm. The second image was recorded 10s after the first one. Scale bars: 200µm. A video visualizing the dynamics during evaporation is attached (S4).**

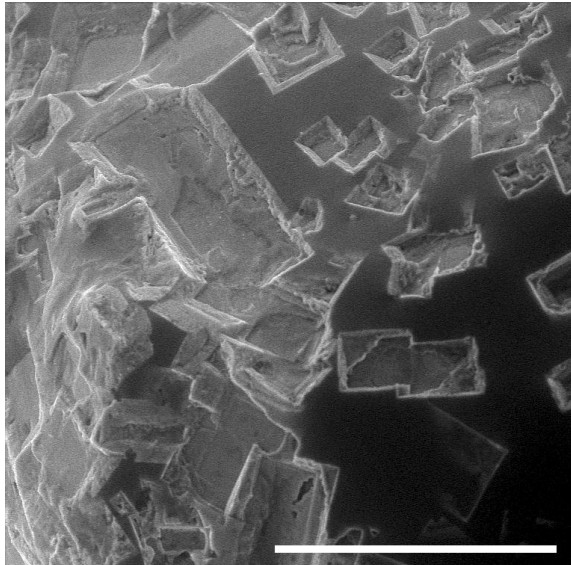

**Figure 8: The NaCl crystals are clearly seen on the top of the surface brine layer of the FF. During the gradual process of evaporation, the individual ice crystals were moving on the brine surface at first, eventually growing into a large cluster of crystals. The ESEM AQUASEM II, ionisation detector, air pressure 510 Pa, sample holder temperature -15.0°C, sample to aperture distance 2mm. Scale bars: 200µm. A video visualizing the formation of these crystals is attached (S5).**

20

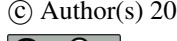



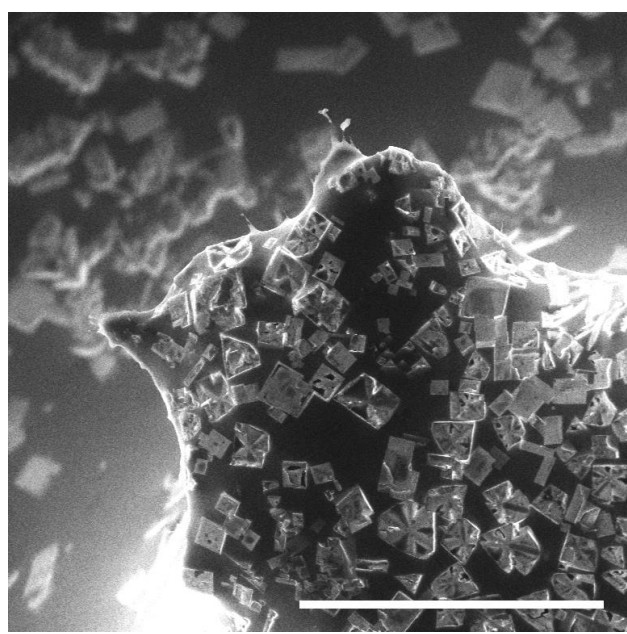

**Figure 9: The NaCl crystal morphologies formed on the FF. The ESEM AQUASEM II, ionisation detector, air pressure 520 Pa, sample holder temperature -15.0°C, sample to aperture distance 2 mm. Scale bar: 200µm.**