# Peer review of "Evaporating brine from frost flowers with electron microscopy, and implications for atmospheric chemistry and sea-salt aerosol formation"

_Atmospheric Chemistry and Physics, 2017_

## Referee Comment (RC1) · Anonymous Referee #1 · 21 Feb 2017

This manuscript, as reflected in the title, addresses relevant scientific questions concerning the role of frost flowers (FF) in sea-salt aerosol (SSA) formation. The novelty of the research mainly relates to the use of an Environmental Scanning Electron Microscope (ESEM) to investigate frost flower dynamics in-situ. The works presents results from experimental temperatures of -5.2 and -17 °C, though further experiments at -10 and -12 °C were completed but not detailed in the manuscript. The most substantial conclusion reached outlines that FF's are not a direct source of SSAs, which is consistent with previous suggestions based on wind tunnel measurements (Roscoe et al., 2011).

The scientific methods and assumptions are clearly outlined, with exception to the use

of a binary NaCl-H2O system in preference to seawater. The experiment descriptions are appropriate for allowing reproduction by fellow scientists. Generally the results are sufficient to support the interpretations, although the discussion of snow particles in section 5.2 is not strictly relevant to the frost flower observations presented. For this reason I suggest removing the first two paragraphs on page 9 (detailed below) and combining sections 5.1 and 5.2. There could also be a greater level of discussion about how, if frost flowers are not a source of SSA's, then why are winter peaks observed with measured [SO42-/Na+] ratios lower than that of seawater?

The authors do provide proper credit to related work with an appropriate quality and quantity of citations and references. The manuscript could benefit from inclusion of more references from literature concerning sea ice brine geochemistry, which would add interdisciplinarity whilst increasing the potential audience.

The abstract provides a concise and complete summary of the manuscript and the presentation throughout continues to be well structured and clear. Generally the language is fluent and precise, however some sentences would benefit from rewording (detailed below).

Mathematical formulae, symbols, abbreviations and units are correctly defined with exception of the use of ANSYS in the caption of figure 2. Lastly, the supplementary materials are an excellent addition to the manuscript. Specific comments

SECTION 1: Introduction

Page 1, line 31: "Outer space" is not relevant to this investigation. Remove.

Page 1, line 34: "Assessing the reactivity" of what?

Page 1, line 40: "Climate index" could instead be "paleo-proxy".

Page 1, line 43: Specify that the "distance to open water" is for polar environments that experience sea ice formation during winter.

Page 2, line 9: It may be beneficial to introduce the source of the brine by describing the sea ice brine system (see Light et al., 2003, Effects of temperature on the microstructure of first-year Arctic sea ice, JGR).

Page 2, line 10: Out of curiosity, why is SO42- removal inapplicable to sea spray? If sea spray is comprised of seawater aerosol particles that are subjected to sub-zero atmospheric conditions, why couldn't mirabilite precipitation occur on microscopic scales? I should stress that the manuscript does not need to be amended in relation to this comment.

Page 2, line 10: Experimental and model evidence has concluded that mirabilite can precipitate from sea ice brine at temperatures $\leq$-6.4 $°$C (see Marion et al., 1999, "Alternative pathways for seawater freezing", Cold Reg. Sci. Tech.; Butler et al., 2016, "Mirabilite solubility in equilibrium sea ice brines", GCA). The original estimate of -8 $°$C comes from Nelson and Thompson (1954, "Deposition of salts from seawater by frigid concentration"), with experiments not given sufficient time to attain equilibrium.

Page 2, line 17: This brine salinity is within the region of mirabilite precipitation shown in Butler et al., (2016, "The effect of mirabilite precipitation on the absolute and practical salinities of sea ice brines", Mar. Chem.).

Page 2, line 20: "high salinity" should be "high brine salinity". Furthermore, this sentence could be reworded with a more explicit description of the chemical reactions being referred to.

SECTION 2: Growth of the frost flowers and preparation of the samples

Page 3, line 5: The use of an aqueous solution of NaCl instead of seawater should be justified, with potential limitation of this method outlined.

Page 3, line 9: Can the quality of figure 1 be improved at all? At the moment it is a little hard to interpret.

Page 3, line 11: Remove "extremely".

SECTION 3: Environmental scanning electron microscope

Sections 2 and 3 both describe the methods of the investigation. I would therefore suggest creating a section 2 titled "Methods", and then including the two current method sections as subsections (i.e. 2.1 and 2.2).

Page 3, line 37: Remove "material".

Page 3, line 41: In the caption of figure 2, what is ANSYS?

SECTION 4: Results This section could be retitled "Results and discussion".

Page 4, line 16: Remove brackets starting after "are".

Page 4, line 18: I have noticed that in some cases a space is used before "°C", and in others there isn't a space. I would recommend including a space throughout.

Page 4, line 19: The way the sentence is worded at the moment makes it difficult to interpret. Is the viscosity of the brine nearly two times higher or two times lower than that of pure water? Also, what do the authors mean by 'ambient' temperature?

Page 4, line 20: Full stop on wrong side of reference.

Page 4, line 32: Amend citation for Buck (1981).

Page 4, line 33: Amend citation.

Page 5, line 6: Should "23.3 %" actually be 8.3 %? Also, with figure 4, I would suggest amending the x and y axis scales so that the experimental conditions of this investigation can be interpreted more easily.

Page 5, line 8: As a general thought, if equilibrium brine concentrations in frost flowers are maintained by ice melting, then the equilibrium composition of the brine within a frost flower should reflect that of sea ice brine inclusions at equilibrium. This just highlights an overlap between the study of frost flowers and sea ice brines.

Page 5, line 36: In relation to changing pH in sub-zero brines, Papadimitriou et al.

(2016, "The measurement of pH in saline and hypersaline media at sub-zero temperatures: Characterization of Tris buffers", Mar. Chem.) or Rerolle et al. (2016 "Measuring pH in the Arctic Ocean: Colorimetric method or SeaFET?", Meth. Ocean.) might be more relevant references.

Page 6, line 6: Explicitly refer to how this description is for experiments at -5 °C.

Page 6, line 18: Rather than "Apparently", it would be better to relate this sentence to the NaCl-H2O phase diagram (figure 4).

Page 6, line 28: The manuscript seems to skip from S3 to S5. Is there an S4?

Page 6, line 36: Butler and Kennedy (2015, "An investigation of mineral dynamics in frozen seawater brines by direct measurement with synchrotron X-ray powder diffraction", JGR: Oceans) would be a more suitable reference for the fast rate of hydrohalite dissolution upon warming.

Page 7, line 3: "thermodynamic conditions" should be "thermodynamic equilibrium conditions".

SECTION 5: Atmospheric implications

I think that this section should become section 4.3, and the current sections 5.1 and 5.2 should become part of this (rather than being separate).

Page 7, line 35: Remove "concrete".

Page 7, line 36: As this investigation is for the "general case of the ice-NaCl system", the manuscript would benefit from a considered discussion about the limitations of using this binary system instead of seawater. According to the composition of standard seawater (Millero et al., 2008, "The composition of Standard Seawater and the definition of the Reference-Composition Salinity Scale", Deep Sea Res.), Na+ and Cl- comprise 85.7 % of the total salt in seawater by mass. The remaining 14.3 % of solutes may play an important role in the geochemistry of frost flowers, particularly since

mirabilite can precipitate from seawater when cooled below -6.4 °C.

Page 8, line 26: "condition" should be "conditions".

Page 8, line, 42: If FFs are not a direct source of SSAs, then what theories are there for the winter maximums in SSA's and the observation of [SO42-/Na+] being lower than seawater? The discussion here would greatly benefit from an appraisal of this.

Page 9, lines 1 – 15: I cannot see how these two paragraphs relate to the current investigation. Surely the study of SSA formation from snow and frost flowers requires two separate investigations? The discussion might be better of concentrating upon the topic outlined in the previous comment.

SECTION 6: Conclusions

Page 9, line 23 – 24: "This microphysical picture applies to not only the high saline FFs, but also the less saline snows, including blown ones". I am not sure how this link is made, and how appropriate it is given that the measurements carried out in this investigation were solely on frost flowers.

---

## Referee Comment (RC2) · Anonymous Referee #2 · 25 Feb 2017

In this work, recent advances in higher-pressure electron microscopy were utilized to observe artificial frost flowers made from aqueous NaCl solutions over a range of temperatures under conditions in which water is evaporating. For the first time the morphology of these surfaces have been observed with micron-scale resolution. This addresses an important and open question about the morphologies of reactive environmental ice surfaces. The images presented in the figures and supplemental videos are detailed, clear, and illuminating. The authors convincingly argue that they are observing brine covering ice at warmer temperatures, and sodium chloride crystals forming on brine covering ice at colder temperatures. They speculate as to some reasonable possible implications for atmospheric chemistry and sea salt aerosol formation. One of the

main conclusions of this work is that frost flowers are robust and sticky and thus won't fall apart to form micrometer-sized particles that could contribute to sea-salt aerosols. This contradicts previous assumptions made about frost flowers, but is corroborated by another recent study. I have two specific questions, followed by a number of technical corrections.

How does the electron beam affect the surface over the time of the experiment? This issue is alluded to in the first paragraph of section 3 (p 3 lines 20-24). The claim is that the electron beam may heat the sample a degree or two. Is this the only effect of the electron beam? Could the temperature gradient be larger for samples at colder temperatures than that at which they observed the ice surface melting? Does the temperature gradient increase over the observation time?

Why is the residual NaCl in Figure S3 not composed of cubic crystals?

Technical Corrections

p 1 line 25: "The present microscopic observation. . ." Replace A with The.

p 2 lines 19-21: This sentence is awkward for a couple reasons. Something like, "The fragile structure plus extremely high salinity make FFs the likely cause of chemical reactions and source for SSA." may express the authors' point better.

p 4 line 31: Adding the phrase, "These values were calculated from the applied equations. . ." would clarify this section.

p 4 lines 34-35: I don't know what the Journal's editorial standards are regarding mathematical formulas, but I would suggest times symbols, $\times$, instead of asterisks in the equations.

p 5 line 33: Add a dash to "freeze-concentrated solution"

p 6 line 1: Add the at the end of the line: ". . .compared to water at the same temp"

p 8 line 5: Replace "placed" with "located."

p 8 line 19: Should read "enhanced bromide liberation" (missing the d on enhanced)

p 9 lines 10-15: The purpose of this paragraph is unclear. Is it to show why SSA is important? It doesn't seem to add anything to the manuscript.

p 13 lin 35: Figure 1 is very hard to see. Can it be improved at all?

The caption for Figure S3 is missing the length of the scale bar.

---

## Author Comment (AC1) · 22 Mar 2017

The response to the referee has following formatting: referee's texts are in black, our responses are in blue, and in italics when we cite from the manuscript.

We are thankful to anonymous referee #1 for constructive scientific, language and typographic comments.

Reply to referee #1 comments:

This manuscript, as reflected in the title, addresses relevant scientific questions concerning the role of frost flowers (FF) in sea-salt aerosol (SSA) formation. The novelty of the research mainly relates to the use of an Environmental Scanning Electron Micro scope (ESEM) to investigate frost flower dynamics in-situ. The works presents results from experimental temperatures of -5.2 and -17C, though further experiments at -10 and -12C were completed but not detailed in the manuscript. The most substantial conclusion reached outlines that FF's are not a direct source of SSAs, which is consistent with previous suggestions based on wind tunnel measurements (Roscoe et al.,2011). The scientific methods and assumptions are clearly outlined, with exception to the use of a binary NaCl-H2O system in preference to seawater. The experiment descriptions are appropriate for allowing reproduction by fellow scientists.

We thank referee #1's general positive comments to our work. However, we do recognize that the referee's major concern is the use of binary NaCl-water instead of the standard seawater. We admit that the frost-flowers prepared from the standard seawater would provide information more directly related to the natural conditions and the suggestion is worth the future experiments. Especially, the precipitation of the mirabilite should occur at the studied temperatures as pointed out by the referee. As suggested in another comment (*Page 7, line 36*), a discussion on the limitation of using this binary system is added in the revision section 4:

*'Apart from NaCl, other salts, which comprise about 14.3% of solutes in seawater by mass [Millero et al., 2008], are also important to the formation and geochemistry of FFs, particularly since mirabilite can precipitate from seawater when temperature is below 6.4 °C. Obviously, the omission of other salts in this study is a limitation to represent real FFs. However, we can surely consider our observations a good model for the general case of the ice-NaCl system.'*

Given the major aim of this study is to investigate whether sea salt aerosol (SSA) could be created through a sublimating process and how many SSA particles will be formed from per single saline crystal, we think the use of pure NaCl salt could provide good estimate of the answer. Thus, the current results should be considered as a model of the real situation.

Generally the results are sufficient to support the interpretations, although the discussion of snow particles in section 5.2 is not strictly relevant to the frost flower observations presented. For this reason I suggest removing the first two paragraphs on page 9 (detailed below) and combining sections 5.1 and 5.2.

We accepted the suggestion and created the section 2. Methods, that consists both 2.1 the frost flowers preparation procedure and 2.2 description of the microscope. Therefore in the revision, old section 5 has been changed into to section 3.3; and old section 5.1 and 5.2 have been changed into section 4.1 and 4.2 respectively. Also we have moved the old last paragraph on page 9 line 10-15 to the new section 4.1 (as its last paragraph, see new text below).

The authors do provide proper credit to related work with an appropriate quality and quantity of citations and references. The manuscript could benefit from inclusion of more references

from literature concerning sea ice brine geochemistry, which would add interdisciplinarity whilst increasing the potential audience.

We add a new paragraph in the revision (see below) to address the brine on sea ice. This is also relevant to your specific comment regarding page 2 line 9.

*'The sea ice microstructure is permeated by brine channels and pockets that contain concentrated seawater-derived brine. Cooling the sea ice results in further formation of pure ice within these pockets as thermal equilibrium is attained, resulting in a smaller volume of increasingly concentrated residual brine (Light et al., 2003;Butler et al., 2016b). A fraction of the concentrated brine will be expelled upwards to form a thin layer of brine on the surface of the sea ice, where FFs can grow under a certain weather condition. The formation of mirabilite in sea ice results in most dissolved $SO_4^{2-}$ being removed from brine, with less effect on $Na^+$ due to the large abundance of sodium relative to the sulphate e.g. (Butler et al., 2016b). SSA produced from these residual brines displays a depleted $[SO_4^{2-}/Na^+]$ ratio as a result. However, for sea spray particles, $Na_2SO_4$ will not be fractionated in the atmosphere or following deposition even though they suffer sub-zero temperatures as the precipitated mirabilite remains within the body of the aerosol and has no effective pathway for the escape.'*

The suggested references were added into the manuscript.

The abstract provides a concise and complete summary of the manuscript and the presentation throughout continues to be well structured and clear. Generally the language is fluent and precise, however some sentences would benefit from rewording (detailed below).
Mathematical formulae, symbols, abbreviations and units are correctly defined with exception of the use of ANSYS in the caption of figure 2. Lastly, the supplementary materials are an excellent addition to the manuscript.

Specific comments:

We thank you for your comments. Responses to your specific comments are shown below.

There could also be a greater level of discussion about how, if frost flowers are not a source of SSA's, then why are winter peaks observed with measured $[SO_4^{2-}/Na^+]$ ratios lower than that of seawater?

In the new revision instruction, a new paragraph (see below) has been added to address blowing snow as a source of SSA proposed, in addition to the FFs:
Apart from saline FFs, snow lying on sea ice can be contaminated by sea water (or saline) through various pathways (Domine et al., 2004). These contaminated salty snows have been hypothesized to act as an efficient source of SSA (via blowing snow) and bromine (Yang et al., 2008;Legrand et al., 2016;Zhao et al., 2016;Levine et al., 2014). The relative importance of these two sea-ice-sourced SSA to the polar winter sea salt budget is still under debate e.g. (Huang and Jaeglé, 2016;Xu et al., 2016;Rhodes et al., 2017). As in any case (FFs or salty snow), the formation of SSA from salty ice particles requires its size to be reduced via the loss of water through either evaporation or sublimation process, depending on the temperature. Until now, there has been no detailed image at the micro-physical scale to indicate what happens to saline ice under the evaporation or sublimation processes.

Page 1, line 31: "Outer space" is not relevant to this investigation. Remove.

Removed.

Page 1, line 34: "Assessing the reactivity" of what?

Of the impurities. Supplemented.

Page 1, line 40: "Climate index" could instead be "paleo-proxy".

Agree. We have changed it into 'paleo-climate proxy'.

Page 1, line 43: Specify that the "distance to open water" is for polar environments that experience sea ice formation during winter.

Thanks for the suggestion, have added 'in polar region' for the sites which experience sea ice formation in winter. Now it reads as: *'the winter SSA peaks observed at most near coastal sites in polar regions (Wagenbach et al., 1998; Rankin et al., 2004) are clearly out of phase with the distance to the open water.'*

Page 2, line 9: It may be beneficial to introduce the source of the brine by describing the sea ice brine system (see Light et al., 2003, Effects of temperature on the microstructure of first-year Arctic sea ice, JGR).

Please see our response to your general comment regarding to the same issue. As can be seen in the revision, a new paragraph has been added in the introduction section to address it.

Page 2, line 10: Out of curiosity, why is SO4$^{2-}$ removal inapplicable to sea spray? If sea spray is comprised of seawater aerosol particles that are subjected to sub-zero atmospheric conditions, why couldn't mirabilite precipitation occur on microscopic scales? I should stress that the manuscript does not need to be amended in relation to this comment.

Please find the new paragraph added in the introduction (also see our response to your general comment). A brief reply: yes, at very low temperature, $Na_2SO_4.10H_2O$ can be formed in sea spray, however, they are kept within the aerosol and there is no effective pathway for the sulphate to escape from the sea salt, thus the original ratio in sea water will mostly be kept in sea spray.

Page 2, line 10: Experimental and model evidence has concluded that mirabilite can precipitate from sea ice brine at temperatures _-6.4 _C (see Marion et al., 1999, "Alternative pathways for seawater freezing", Cold Reg. Sci. Tech.; Butler et al., 2016, "Mirabilite solubility in equilibrium sea ice brines", GCA). The original estimate of -8 °C comes from Nelson and Thompson (1954, "Deposition of salts from seawater by frigid concentration"), with experiments not given sufficient time to attain equilibrium.

Page 2, line 17: This brine salinity is within the region of mirabilite precipitation shown in Butler et al., (2016, "The effect of mirabilite precipitation on the absolute and practical salinities of sea ice brines", Mar. Chem.).

Thank you for the correction. The value of -8 °C was substituted by -6.4 °C and the appropriate reference was added.

Page 2, line 17: This brine salinity is within the region of mirabilite precipitation shown in Butler et al., (2016, "The effect of mirabilite precipitation on the absolute and practical salinities of sea ice brines", Mar. Chem.).

We have added words to mention that. Now it reads as: *'...FFs can reach the salinity of the concentrated brine of 120 practical salinity units (Douglas et al., 2012), which is in the range of the mirabilite precipitation (Butler et al., 2016b).'*

Page 2, line 20: "high salinity" should be "high brine salinity". Furthermore, this sentence could be reworded with a more explicit description of the chemical reactions being referred to.
The sentence was reformulated (as suggested by Referee 2), the word "brine" was added and reactivity was specified. Now the sentence reads:
'*The fragile structure plus extremely high brine salinity (Rankin, 2002) make FFs the likely cause of chemical reactions (e.g. heterogenous, photochemical, and redox) (Perovich and Richter-Menge, 1994;Kaleschke et al., 2004;Simpson et al., 2007) and source for SSA (Wagenbach et al., 1998;Wolff et al., 2003).* '

Page 3, line 5: The use of an aqueous solution of NaCl instead of seawater should be justified, with potential limitation of this method outlined.
As mentioned above in the answer to your general comment regarding the same question, the main aim of this study was to investigate whether evaporating or sublimating salty crystals could cause sea salt aerosol (NaCl) formation and how many sea salt aerosol could generated from one single crystal. Thus, the large amount of various impurities in real sea water could prevent us from deriving a robust conclusion. Nevertheless, we recognize the importance of the experiments with the seawater and consider them for the future.

Page 3, line 9: Can the quality of figure 1 be improved at all? At the moment it is a little hard to interpret.
The contrast of the picture was increased, resulting in more readable picture.

Page 3, line 11: Remove "extremely".
Removed.

Sections 2 and 3 both describe the methods of the investigation. I would therefore suggest creating a section 2 titled "Methods", and then including the two current method sections as subsections (i.e. 2.1 and 2.2).
The suggestion was accepted and the chapters were rearranged accordingly.

Page 3, line 37: Remove "material".
We think the word „material" is appropriate in this context, as we wanted to emphasize that the detector of back-scattered electrons is material sensitive.

Page 3, line 41: In the caption of figure 2, what is ANSYS?
The information about the software was added: **The ANSYS Inc.© Fluent.**

SECTION 4: Results This section could be retitled "Results and discussion".
The suggested title was accepted.

Page 4, line 16: Remove brackets starting after "are".
Removed.

Page 4, line 18: I have noticed that in some cases a space is used before "°C", and in others there isn't a space. I would recommend including a space throughout.
Supplied. Thank you for noting the inconsistencies.

Page 4, line 19: The way the sentence is worded at the moment makes it difficult to interpret. Is the viscosity of the brine nearly two times higher or two times lower than that of pure water? Also, what do the authors mean by 'ambient' temperature?

The direction of viscosity change was added and the temperature was specified.

Page 4, line 20: Full stop on wrong side of reference.
Amended.

Page 4, line 32: Amend citation for Buck (1981).
Done.

Page 5, line 6: Should "23.3 %" actually be 8.3 %? Also, with figure 4, I would suggest amending the x and y axis scales so that the experimental conditions of this investigation can be interpreted more easily.
The referee is right. Thank you for correcting the error.

Page 5, line 8: As a general thought, if equilibrium brine concentrations in frost flowers are maintained by ice melting, then the equilibrium composition of the brine within a frost flower should reflect that of sea ice brine inclusions at equilibrium. This just highlights an overlap between the study of frost flowers and sea ice brines.
Yes, we agree that the brine within sea ice should behave in very similar manner to that of frost flower.

Page 5, line 36: In relation to changing pH in sub-zero brines, Papadimitriou et al. (2016, "The measurement of pH in saline and hypersaline media at sub-zero temperatures: Characterization of Tris buffers", Mar. Chem.) or Rerolle et al. (2016 "Measuring pH in the Arctic Ocean: Colorimetric method or SeaFET?", Meth. Ocean.) might be more relevant references.
The references were added. We consider the freezing and subsequent ions redistribution the more important factors then the temperature alone and the method of applying the spectroscopic of indicator dye as more appropriate estimate for the acidity in concentrated brine solution on the surfaces of ice.

Page 6, line 6: Explicitly refer to how this description is for experiments at -5 °C.
We moved this paragraph to the end of the section 3.2, which deals with temperatures of -17.2 C (where it really belongs to). The trend at -5 C is similar only less pronounced, therefore we do not consider important to discuss it separately.

Page 6, line 18: Rather than "Apparently", it would be better to relate this sentence to the NaCl-H2O phase diagram (figure 4).
We linked the sentence to the phase diagram. However, we left the "Apparently" statement to emphasize the observation is needed for the conclusions.

Page 6, line 28: The manuscript seems to skip from S3 to S5. Is there an S4?
The reference to S4 was added to Figure 7. Thank you for noting it.

Page 6, line 36: Butler and Kennedy (2015, "An investigation of mineral dynamics in frozen seawater brines by direct measurement with synchrotron X-ray powder diffraction", JGR: Oceans) would be a more suitable reference for the fast rate of hydrohalite dissolution upon warming.
The reference was switched.

Page 7, line 3: "thermodynamic conditions" should be "thermodynamic equilibrium conditions".
The term: equilibrium, was added.

SECTION 5: Atmospheric implications
I think that this section should become section 4.3, and the current sections 5.1 and 5.2 should become part of this (rather than being separate).
The new order has been sorted out in the revision.

Page 7, line 35: Remove "concrete".
Ok.

Page 7, line 36: As this investigation is for the "general case of the ice-NaCl system", the manuscript would benefit from a considered discussion about the limitations of using this binary system instead of seawater. According to the composition of standard seawater (Millero et al., 2008, "The composition of Standard Seawater and the definition of the Reference-Composition Salinity Scale", Deep Sea Res.), Na+ and Cl- comprise 85.7 % of the total salt in seawater by mass. The remaining 14.3 % of solutes may play an important role in the geochemistry of frost flowers, particularly since mirabilite can precipitate from seawater when cooled below -6.4 ₒC.

We fully agree with the referee's concern of not using standard seawater compounds to represent the nature seawater. Thus, the limitation of the simple binary $NaCl$-$H_2O$ systems is clear, as addressed in reply to referee's general comments on page 1. In the revision section 4, we add a sentence to discuss this limitation of it:
*'It should be stressed that $Na^+$ and $Cl^-$ comprise 85.7 % of the total salt in seawater by mass (Millero et al., 2008). The remaining 14.3 % of solutes may play an important role in the geochemistry of FFs or the general ice with sea salt, and thus may be important for certain considerations. Especially the precipitated ikaite and mirabilite from the sea water, at -2 °C and -6.4 °C respectively, may possibly act as nucleation centres for NaCl (Butler et al., 2016a). Obviously, the omission of other salts in this study is a limitation to represent real FFs.'*

Page 8, line 26: "condition" should be "conditions".
Ok.

Page 8, line, 42: If FFs are not a direct source of SSAs, then what theories are there for the winter maximums in SSA's and the observation of [SO42-/Na+] being lower than seawater? The discussion here would greatly benefit from an appraisal of this.

We are grateful for encouragement to start the discussion of this topic. In the conclusion, we add new sentences:
*'As hypothesized by Yang et al. (2008), that blown salty snow particle on sea ice, via a sublimation process, could act as an efficient SSA source. Recent modelling studies further support this assumption, as observed winter SSA peaks in most polar costal sites and inland sites can be well reproduced by modelling (Levine et al., 2014;Huang and Jaeglé, 2016;Rhodes et al., 2017).'*

Page 9, lines 1 – 15: I cannot see how these two paragraphs relate to the current investigation. Surely the study of SSA formation from snow and frost flowers requires two separate investigations? The discussion might be better of concentrating upon the topic outlined in the previous comment.

We fully understand referee's concern, however, we think the discussion on the salty snow as a –source of SSA, via sublimation process is useful, particularly the part on the ratio of how many sub-SSA that could be formed from sublimating one single snow particle. As the conclusion derived (e.g. the ratio is taken as a unit) is completely based on the observation taken in this lab work (as the brine fingers are quite sticky and hard to break), a similar phenomenon that could occur in less salty snow particle. Certainly, this conclusion needs further investigation to confirm, but we suppose they should behave in similar manner. For this reason, we keep the discussion on salty snow in the revision, but slightly adjust the wording. Also we have moved the old last paragraph (on P9 line 10-15) into the new section 4.1 as its last paragraph.

SECTION 6: Conclusions
Page 9, line 23 – 24: "This microphysical picture applies to not only the high saline FFs, but also the less saline snows, including blown ones". I am not sure how this link is made, and how appropriate it is given that the measurements carried out in this investigation were solely on frost flowers.
We agree with referee's comment and now we have deleted this sentence.

**Reference**

[revised manuscript text omitted]

---

## Author Comment (AC2) · 22 Mar 2017

We are thankful to anonymous referee #2 for constructive scientific, language and typographic comments.

**Reply to referee #2 comments:**

How does the electron beam affect the surface over the time of the experiment? This issue is alluded to in the first paragraph of section 3 (p 3 lines 20-24). The claim is that the electron beam may heat the sample a degree or two. Is this the only effect of the electron beam? Could the temperature gradient be larger for samples at colder temperatures than that at which they observed the ice surface melting? Does the temperature gradient increase over the observation time?

There is a cumulative effect of the electron beam with the energy of 20 keV and of the ions produced inside the ESEM chamber due to the interaction of electrons with a gas. The largest number of ions is generated at the distance of 1 mm from the detection electrode, where the intensity of the field is ≤ 200 V. For that reason, there is only mild effect of the ions on the ice surface, and the heating and disruption of the ice surface by the ions is almost negligible in our ESEM. The applied ESEM contrasts to a microscope with ion beam, where ions are focused to one place and their energy during the collision with the sample surface is generally more than 1 kV.

As the bottom of the ice is cooled by Peltier cooler and the surface is heated by the electron beam, temperature gradient is expected to slightly rise in time. There are two antagonistic effects when considering the dependence of the temperature gradient on temperature. The thermal conductivity of ice is inversely proportional to temperature (Rabin, 2000); therefore, the temperature gradient should be smaller at the lower temperature. On the other hand, water vapour with the temperature around 0 °C is blown to the microscope chamber near the sample surface. This could increase the temperature gradient at lower temperatures. Therefore, it is difficult to predict the overall dependence of the temperature gradient on temperature under given experimental conditions. Except for the observation of the ice melting temperature, we do not have any mean to estimate the sample temperature. However, the sample is relatively stable in time during the experiments if evaporation and condensation are avoided.

Heating of the sample by electron beam, the temperature gradients, and the surface disruption are substantially lower in our non-commercial microscope in comparison with common commercial ESEMs. In our microscope, electron flux is four times lower compared to the ones used in common ESEMs, and the radiation damage is decreased because of high scan rate. One of the fundamental differences between ESEM and SEM is that the electron beam is scattered by the gas in the chamber before it collides with the sample.

Why is the residual NaCl in Figure S3 not composed of cubic crystals?

We suppose that the macroscopic arrangement of the lyophile is a result of evaporation process which did not allow enough time for formation of large NaCl crystals. Also the presence of residual water can contribute to the final appearance of the lyophile.

Technical Corrections
p 1 line 25: "The present microscopic observation: : :" Replace A with The.
p 2 lines 19-21: This sentence is awkward for a couple reasons. Something like, "The fragile structure plus extremely high salinity make FFs the likely cause of chemical reactions and source for SSA." may express the authors' point better.
p 4 line 31: Adding the phrase, "These values were calculated from the applied equations: : :" would clarify this section.

p 4 lines 34-35: I don't know what the Journal's editorial standards are regarding mathematical formulas, but I would suggest times symbols, _, instead of asterisks in the equations.
p 5 line 33: Add a dash to "freeze-concentrated solution"
p 6 line 1: Add the at the end of the line: ": : :compared to water at the same temp"
p 8 line 5: Replace "placed" with "located."

p 8 line 19: Should read "enhanced bromide liberation" (missing the d on enhanced)

Thanks to the Referee 2 for corrections – they were all incorporated.

p 9 lines 10-15: The purpose of this paragraph is unclear. Is it to show why SSA is important? It doesn't seem to add anything to the manuscript.

The purpose of this paragraph is to address the chemical potential of sea salt produced via the sublimation process. This paragraph was modified and now moved to section 4.1.

p 13 lin 35: Figure 1 is very hard to see. Can it be improved at all?

We increased the contrast of this picture.

The caption for Figure S3 is missing the length of the scale bar.

The missing length of the scale bar was added.

References:

Rabin, Y.: The effect of temperature-dependent thermal conductivity in heat transfer simulations of frozen biomaterials, Cryo-Letters, 21, 163-170, 2000.